# ChronobioticsDB: The Database of Drugs and Compounds Modulating Circadian Rhythms

**DOI:** 10.3390/clockssleep7030030

**Published:** 2025-06-23

**Authors:** Ilya A. Solovev, Denis A. Golubev, Arina I. Yagovkina, Nadezhda O. Kotelina

**Affiliations:** Laboratory of Translational Bioinformatics and Systems Biology, Medical Institute, Pitirim Sorokin Syktyvkar State University, Oktyabrsky Prosp. 55, 167000 Syktyvkar, Russia; denismeatboy@gmail.com (D.A.G.); zedgolis@mail.ru (A.I.Y.); nkotelina@gmail.com (N.O.K.)

**Keywords:** database, chronobiotics, circadian rhythm modulators, clock modulators, core clock pharmacology, chronodisruptors, drugs repurposing, chronomedicine

## Abstract

Chronobiotics represent a pharmacologically diverse group of substances, encompassing both experimental compounds and those utilized in clinical practice, which possess the capacity to modulate the parameters of circadian rhythms. These substances influence fluctuations in various physiological and biochemical processes, including the expression of core “clock” genes in model organisms and cell cultures, as well as the expression of clock-controlled genes. Despite their chemical heterogeneity, chronobiotics share the common ability to alter circadian dynamics. The concept of chronobiotic drugs has been recognized for over five decades, dating back to the discovery and detailed clinical characterization of the hormone melatonin. However, the field remains fragmented, lacking a unified classification system for these pharmacological agents. The current categorizations include natural chrononutrients, synthetic targeted circadian rhythm modulators, hypnotics, and chronobiotic hormones, yet no comprehensive repository of knowledge on chronobiotics exists. Addressing this gap, the development of the world’s first curated and continuously updated database of chronobiotic drugs—circadian rhythm modulators—accessible via the global Internet, represents a critical and timely objective for the fields of chronobiology, chronomedicine, and pharmacoinformatics/bioinformatics. The primary objective of this study is to construct a relational database, ChronobioticsDB, utilizing the Django framework and PostGreSQL as the database management system. The database will be accessible through a dedicated web interface and will be filled in with data on chronobiotics extracted and manually annotated from PubMed, Google Scholar, Scopus, and Web of Science articles. Each entry in the database will comprise a detailed compound card, featuring links to primary data sources, a molecular structure image, the compound’s chemical formula in machine-readable SMILES format, and its name according to IUPAC nomenclature. To enhance the depth and accuracy of the information, the database will be synchronized with external repositories such as ChemSpider, DrugBank, Chembl, ChEBI, Engage, UniProt, and PubChem. This integration will ensure the inclusion of up-to-date and comprehensive data on each chronobiotic. Furthermore, the biological and pharmacological relevance of the database will be augmented through synchronization with additional resources, including the FDA. In cases of overlapping data, compound cards will highlight the unique properties of each chronobiotic, thereby providing a robust and multifaceted resource for researchers and practitioners in the field.

## 1. Introduction

The regulation of biological rhythms is a fundamental aspect of human physiology, linked to the concept of circadian rhythms, which govern a wide array of bodily functions over a roughly 24 h cycle. These rhythms are driven by the circadian clock, a complex network of genes and proteins that modulate key processes such as sleep–wake cycles, hormonal secretion, metabolic functions, and cognitive performance. Disruptions to the circadian clock, whether due to environmental factors, lifestyle choices, or underlying health issues, can lead to a variety of adverse health outcomes, including sleep disorders, metabolic syndrome, and mood disturbances [1].

In recent years, a growing body of literature has focused on the role of pharmacological agents, referred to as chronobiotics, in the modulation of the circadian clock. Chronobiotics can be defined as drugs or natural compounds that promote the synchronization of biological rhythms, enhancing the alignment of physiological processes with environmental cues. These agents have shown promise not only in the treatment of circadian rhythm disorders, such as delayed sleep phase disorder, but also in addressing broader health challenges associated with circadian misalignment. The identification and characterization of chronobiotic compounds necessitate a systematic approach to cataloging the existing research, elucidating their mechanisms of action, and assessing their clinical utility. This article aims to provide a comprehensive database of drugs and compounds with chronobiotic properties, detailing their pharmacological profiles, therapeutic applications, and potential side effects. By compiling this information, we seek to establish a valuable resource for researchers and clinicians alike, facilitating the translation of chronobiological principles into effective therapeutic strategies [2].

Moreover, understanding the symbiotic relationship between circadian rhythms and pharmacology has implications that extend beyond isolated medical conditions. As society grapples with the increasing prevalence of shift work, frequent travel across time zones, and lifestyle factors that contribute to circadian disruption, the need for effective chronobiotic therapies becomes ever more pressing [3].

This database not only serves as a reference point for the existing knowledge but also aims to inspire future research that could further elucidate the role of circadian modulation in health and disease. The exploration of chronobiotics represents a promising frontier in the intersection of chronobiology and pharmacology. This article endeavors to contribute to the growing body of knowledge in this field, offering a structured overview of compounds capable of influencing the circadian clock and their implications for future clinical applications on different levels of life organization [4].

## 2. Results

### 2.1. Database Web Interface

ChronobioticDB (accessible at https://chronobiotic.ru, access date 16 June 2025) is an online database designed to provide a centralized resource on chronobiotic compounds (drugs or molecules that modulate circadian rhythms), addressing the lack of a unified knowledge base in this domain.The web interface is organized into two complementary components that together support both broad data dissemination and expert curation (Figure 1). The publicly accessible front-end displays a catalog of known and candidate chronobiotics in a searchable, filterable table, enabling users to quickly find entries by compound name, molecular formula, or chemical structure (SMILES notation). Each entry is summarized with key fields such as the compound’s name, molecular formula, and FDA approval status, and users can click an entry to view additional details (e.g., a descriptive synopsis and reference links) in an intuitive pop-up panel. This user-friendly interface allows researchers and general users to easily explore the database and retrieve information, thereby promoting accessibility and dissemination of chronobiotic-related data (Figure 2).

A dedicated and user-friendly submission form is available for researchers working on chronobiotics, which can be completed upon receiving administrative approval. This provision is applicable to both researchers who have published data on novel compounds and those who have repurposed existing registered substances for use as chronobiotics after translational research.

Co-authors contributing to the database will be afforded the opportunity to provide a comprehensive description of the compounds, including the option to present their chemical formulas in the SMILES format. Additionally, they will be able to supply references to other sources where the compound has been previously discussed or described. There exists the possibility of including a concise textual annotation regarding the properties of the compound within the context of chronobiology, thereby emphasizing its potential influence on biological rhythms.

It is important to note that all the submissions containing information about the compounds will be subject to a moderation process. This procedure is designed to ensure the high quality and relevance of the information provided, which is of paramount importance for advancing research in this domain.

### 2.2. Database Statistics and Analysis

In this section we tried to analyze the percentage of different compounds in ChronobioticsDB (Figure 3). The pie chart (Figure 1) illustrates the distribution of chronobiotic compounds categorized by their respective drug classes. The largest proportion of compounds (18%) are CRY ligands, followed by steroids (13%) and melatonin receptor agonists (12%). Anesthesia drugs account for 10% of the compounds, while FDA-approved drugs represent 9% [5]. Natural products, classified as chrononutrients, constitute 5%, the same proportion as ORX1,2R ligands. Antibiotics with chronobiotic properties and BMAIL1 ligands each make up 4% of the total. ROR ligands, GSK ligands, and antifungals each represent 3%. Antipsychotics, sirtuin modulators, and REV-ERB ligands each account for 2%. Neuropeptides (EY1 and Y2), PRO-PRE-BIOTICS-CHRONOBIOTICS, and other minor categories (e.g., ALKAIOIDS, OPSIN ligands, Calmodulin modulators, ADORA2B ligands, p38, and PRR) each represent 1% or less, with several categories showing very small representation. This distribution highlights the diversity of chronobiotic compounds and their varying prevalence across different pharmacological classes.

The structure of ChronobioticsDB mirrors this distribution: each compound record is linked to one or more pharmacological classes through the dedicated Bioclassification table, enabling class-level queries and analytics. The total number of links and citations is growing constantly; thus, we tried to give basic statistics on links and objects according to the last updates (Figure 3).

### 2.3. The Classification of Molecules According to the Major Chemical Groups

A well-structured chemical classification system is essential for organizing the database, facilitating drug discovery, predicting pharmacological properties, and understanding structure–activity relationships (SAR). Compounds within the same chemical class often exhibit similar biological activity due to shared functional groups and scaffolds. Classification enables efficient filtering as the information about the class may be retrieved by AI agents later from descriptions.

The SMILES representation of the database can be classified into various chemical classes based on their structural features and functional groups. Here is a systematic classification of ChronobioticsDB content [6]:
Terpenes/Steroids;Alkaloids;Sulfonamides/Sulfonates;Aromatic compounds (Phenols and Flavonoids);Organofluorines/Organochlorines;Quaternary ammonium compounds;Amides/Peptides;Esters/Lipids;Heterocycles (Pyridine and Thiophene);Nitro compounds;Phosphates/Nucleotides;Organometallics/Salts;Macrocyclic compounds;Disulfides;Alkyne derivatives;Barbiturates;Metal salts.


### 2.4. Key Observations on the Proposed Classes in ChronobioticsDB

Terpenes/Steroids are highly lipophilic, often interacting with membrane-bound receptors (e.g., steroid hormones). Alkaloids have basic nitrogen atoms influencing pharmacokinetics (e.g., blood–brain barrier penetration). Subclasses (indoles, tropanes, and isoquinolines) may correlate with specific bioactivities. Sulfonamides/Sulfonates are the key in antibacterials (sulfa drugs) and carbonic anhydrase inhibitors, and these drugs may trigger hypersensitivity reactions in some patients. Organohalogens often have enhanced metabolic stability (e.g., fluoxetine and haloperidol). Chlorinated aromatics may raise toxicity concerns (e.g., bioaccumulation). Quaternary ammonium compounds have a charge that limits absorption but enhances antimicrobial activity. Macrocyclic compounds have constrained structures improving target binding (e.g., cyclosporine and macrolide antibiotics).

Some drugs in ChronobioticsDB belong to multiple categories (e.g., alkaloids with terpene moieties). Hybrid classifications may be needed. New synthetic chemotypes (e.g., PROTACs and covalent inhibitors) may require additional classes and it is the dominating problem of this approach to content classification. A group of circadian rhythm modulators have the same targets and totally different chemical classifications.

Practical implementation in a database may be based on Hierarchical Taxonomy: Broad classes (e.g., “Alkaloids”) → subclasses (e.g., “Indole Alkaloids”), and SMILES-based queries like automatic classification using structural fingerprints which will be incorporated in later versions. The classification will later help in broad activity annotations, linking chemical classes to typical therapeutic uses (e.g., Barbiturates → Sedatives).

A chemically informed classification system enhances drug database utility by enabling structure-based searches, SAR analysis, and predictive modeling. Future refinements could incorporate machine learning SAR/QSAR models for automated class assignments based on emerging chemotypes.

## 3. Discussion

The development of ChronobioticsDB represents a significant advancement in chronobiology, pharmacoinformatics, and bioinformatics, as at the present day only circadian gene expression databases exist [7,8]. Collecting together critical data on compounds (Figure 4) that modulate circadian rhythms, this database addresses a longstanding gap in the scientific community’s ability to systematically study and apply chronobiotics. Through a balanced combination of primary data—manually curated and verified—and secondary data drawn from sources like PubChem [9], DrugBank [10], and ChemSpider [11], ChronobioticsDB not only functions as a repository but also evolves into a dynamic tool for researchers and clinicians.

A core challenge in chronobiology has been the absence of a unified framework for classifying chronobiotic compounds; historically, researchers have had to rely on fragmented resources defined by chemical properties, mechanisms of action, or therapeutic applications. By establishing a relational database structure and organizing compounds according to their chemical, pharmacological, and biological characteristics, ChronobioticsDB enables researchers to detect patterns that might otherwise go unnoticed and to discover entirely new chronobiotics. The use of SMILES notation for chemical structures and the integration of data from multiple sources—ranging from FDA approval statuses to clinical trial results—further bridge the gap between basic research and clinical applications, facilitating the exploration of molecular mechanisms underlying circadian rhythms. This broad view is especially relevant to chronomedicine and therapeutics, as circadian disruption is increasingly linked to metabolic syndromes, neurodegenerative conditions, and mood disorders; by classifying compounds into groups such as CRY ligands, melatonin receptor agonists, and steroids, the database provides a crucial resource for focusing on targeted interventions.

Moreover, synchronizing compound profiles with clinical trial data reveals insights into the safety and efficacy of chronobiotic therapies, helping refine the existing treatments or guide new ones. Despite these innovations, ChronobioticsDB faces important challenges, notably the need for perpetual updates as new findings emerge and the desirability of integrating genomic, transcriptomic, and proteomic data to achieve a richer, system-level view of how these compounds operate on circadian pathways. Adding visualization and analytics tools, such as network mapping or predictive modeling, could further illuminate interactions among compounds, targets, and biological processes, promoting deeper hypothesis generation and testing.

The current version of ChronobioticsDB has some limitations. It contains only the chronobiotic compounds described in published studies, so newly discovered or proprietary modulators are not yet represented. As with other curated databases, the information for each entry is limited by what is reported in the literature, and some experimental details may be incomplete if not provided by the source. Future updates will address these gaps by incorporating new data as it becomes available.

### 3.1. Ethical Considerations

This study did not involve any new experiments with human participants or animals, as ChronobioticsDB was compiled exclusively from previously published data. All the source studies adhered to ethical standards as documented in their original publications. By aggregating this information, we introduce no new ethical concerns. Furthermore, we have ensured that each entry in the database is accompanied by references to the original source, giving due credit and enabling users to verify details in the context of the ethical frameworks of those studies. We are committed to maintaining transparency and responsibility in how data are collected and shared via ChronobioticsDB.

### 3.2. Future Updates and Community Engagement

ChronobioticsDB is intended to be a living resource. We plan to update the database regularly (at least annually or whenever a significant number of new chronobiotic findings emerge in the literature) so that it remains up-to-date. Each update will be documented and versioned. To maximize transparency and collaboration, we will communicate these updates through international scientific associations and forums. For example, we will share news of ChronobioticsDB enhancements via the newsletters and conferences of chronobiology and pharmacology societies, and we invite researchers to contribute by suggesting new compounds or correcting entries. By leveraging international networks and associations, we aim to cultivate a global user community that both benefits from and contributes to ChronobioticsDB.

### 3.3. Limitations, International Validation

We acknowledge that ChronobioticsDB has been initially curated by the authors and has not yet been vetted through an international peer-review process. To enhance its validity and global relevance, we plan to seek feedback from the international chronobiology community. In future updates, we will collaborate with experts worldwide—for example, by forming an international advisory board and inviting contributions or critiques—to ensure the database’s content is rigorously reviewed and improved by a broad range of specialists.

Finally, ethical and practical considerations mandate close attention to intellectual property rights and data security: providing precise citations helps ensure proper attribution, while role-based access and encryption maintain the confidentiality and integrity of the database’s contents. By harmonizing these elements, ChronobioticsDB stands poised to guide both basic chronobiology and clinical innovation, offering a unifying, data-driven platform for the scientific community’s ongoing exploration of circadian rhythm modulation.

## 4. Materials and Methods

### 4.1. Data Acquisition

Information on compound identity, structure, mechanism of action, effect, targets, and observed outcomes was manually extracted.

Quality Assessment: Each study underwent a thorough quality assessment using the established criteria to evaluate study design, sample size, reproducibility of results, etc. (Table 1).

#### 4.1.1. Data Sources and Extraction

The Chronobiotics Database consolidates findings from numerous studies on the influence of chronobiotics on biological rhythms. The database currently comprises a comprehensive collection of more than 900+ unique scientific references, including peer-reviewed journal articles, preclinical trials, and experimental reports and datasets [12]. These sources were curated through systematic searches in established scientific repositories such as PubMed and PubChem, Medline, Mendeley data, ACS Publications, SpringerLink, Google Scholar, Scopus (https://scholar.google.com/, access date 16 June 2025), and Web of Science using keywords related to circadian rhythms, aging, longevity, and chronobiotics. The origin of the present research is gerontological; primarily, the chronobiotics descriptions were archived and categorized by our team for aging desynchronosis treatment in model animals for translational purposes, but while accumulating data we found it interesting to widen the project on all fields of biological rhythm research, but still several modulators of circadian rhythms were described majorly in the framework of geriatric and biogerontology research like resveratrol, mTOR inhibitors, and K185. The desynchronosis is one of the key signs of aging and ChronobioticsDB was created to discover new chronobiotics for its treatments as well as geroprotectors with chronobiotic effects.

Keyword list: chronobiotics, chronodisruptors, circadian rhythm modulators, circadian clock modulators, plant chronobiotics, plant-derived chronobiotics, probiotic chronobiotics, geroprotective chronobiotics, geroprotectors affecting sleep, geroprotectors affecting circadian rhtyhms, circadian rhythm modulators, sleep–wake cycle regulators, biological clock entrainment drugs, circadian phase shift drugs, endogenous pacemakers pharmacological modulators, suprachiasmatic nucleus (SCN) pharmacology, chemical zeitgebers, Photoperiodism affecting drugs, melatonin agonists, melatonin receptor agonists, orexin receptor antagonists, Non-benzodiazepine hypnotics (Z-drugs), chronopharmacology, delayed sleep–wake phase disorder (DSWPD) treatment, advanced sleep–wake phase disorder (ASWPD) treatment, irregular sleep–wake rhythm disorder treatment, non-24 h sleep–wake disorder (Non-24) treatment, shift work disorder treatment, jet lag disorder treatment, circadian misalignment treatment, Social jet lag treatment, free-running disorder treatment, age-related circadian disruption treatment, Senescent circadian decline treatment, elderly sleep fragmentation treatment, sundowning syndrome treatment, Alzheimer’s circadian dysfunction treatment, Parkinson’s sleep disturbances treatment, menopause-related insomnia treatment, sarcopenia circadian links treatment, age-dependent melatonin reduction treatment, vascular aging circadian effects treatment, Insomnia disorder treatment, Obstructive sleep apnea (OSA) treatment, central sleep apnea pharmacological treatment, narcolepsy treatment, hypersomnia treatment, restless legs syndrome (RLS) treatment, REM sleep behavior disorder (RBD) treatment, parasomnias treatment, fatal familial insomnia treatment, CLOCK gene expression, BMAL1 regulation drugs, PER/CRY feedback loop drugs, melatonin synthesis pathway drugs, adenosine signaling drugs, GABAergic sleep regulation drugs, Orexin/hypocretin system drugs, Core body temperature rhythm drugs, cortisol circadian secretion drugs, chronotherapy drugs, sleep restriction therapy drugs, Timed melatonin supplementation, sleep pharmacology, hypnotic drugs, phase advance pharmacological protocols, phase delay pharmacological protocols, phase angle deviation, Pediatric circadian disorders pharmacology, adolescent sleep phase delay pharmacology, menstrual cycle sleep effects pharmacology, pregnancy-related insomnia pharmacology, postpartum circadian disruption pharmacology, dementia-related sleep loss pharmacology, critical illness circadian disruption pharmacology, blindness-associated non-24 pharmacology, shift work scheduling pharmacology, Transmeridian travel pharmacology, social rhythm disruption pharmacological treatment, night-eating syndrome drugs, caffeine chronotoxicity, caffeine pharmacology, chronotoxicants, waking drugs, MT1 melatonin receptors agonists/antagonists, MT2 melatonin receptors agonists/antagonists, orexin receptor OX1R/OX2R agonists/antagonists, GABA A receptors agonists/antagonists, adenosine A1/A2A receptors agonists/antagonists, casein kinase 1ε/δ agonists/antagonists, REV-ERBα agonists/antagonists, RORα modulators/ligands, CRY modulators/ligands, CLOCK modulators/ligands, PER modulators/ligands, BMAL modulators/ligands, Sleep latency reduction, Sleep efficiency improvement, chrononutrition, chrononutrients, Gut microbiome circadian interactions, time-restricted eating (TRE) mimetics, mitochondrial circadian rhythms modulators, epigenetic drugs modulating circadian clock, senolytics for circadian repair, cancer chronotherapy, depression circadian links pharmacology, Bipolar disorder rhythm instability treatment, metabolic syndrome desynchrony treatment, cardiovascular circadian disruption treatment, immunosenescence circadian pharmacology, neurodegenerative timing deficits, and non-canonical circadian modulators.

Data extracted from these sources include primary details, such as compound structures, mechanisms of action, and effects on circadian rhythms, as well as secondary data like experimental conditions and outcomes. This comprehensive and systematic dataset forms the backbone of the ChronobioticsDB, facilitating a deeper understanding of chronobiotic compounds and their potential for circadian rhythm modulation.

#### 4.1.2. Data Quality Assessment and Validation

The data curation process involved several essential stages, each designed to ensure the accuracy, consistency, and ethical integrity of the collected information [13].

A systematic approach was employed to retrieve relevant studies focusing on the effects of various compounds on circadian rhythms. The inclusion criteria for studies were as follows:(a)Peer-reviewed articles published in the past five decades.(b)Research demonstrating both acute and chronic effects on circadian rhythms.(c)Compounds studied through both in vitro and in vivo methodologies.

Detailed information was extracted for each compound, including its identity, structure, mechanism of action, biological targets, effects, and observed outcomes. Each study underwent a rigorous quality assessment based on study design, sample size, and reproducibility of results to ensure reliability.

Cross-validation of the extracted data was performed by comparing the results across multiple studies to confirm accuracy and consistency if the study mentioned is not the only one existing.

#### 4.1.3. Data Integration and Standardization

To facilitate seamless data integration, standardization protocols were employed:(a)Compound names were normalized following the IUPAC nomenclature (if available).(b)Controlled vocabularies were established for biological terms relating to chronobiotic effects to ensure uniformity across entries.

The curation process adhered to ethical guidelines, ensuring respect for intellectual property rights. Proper citations of sources and links to original publications were provided to uphold academic integrity.

#### 4.1.4. Database Implementation

The curated data was integrated into a relational database management system, designed to support easy queries and data retrieval. The database is accessible via a user-friendly web interface, allowing users to search by compound characteristics, effects, and study parameters.

### 4.2. Database Organization (Primary and Secondary Data)

The difference between primary and secondary data in the development of ChronobioticsDB is major for ensuring accuracy and comprehensiveness. Primary data consists of information created specifically for the database, often through manual curation by expert researchers. This includes molecular structures (e.g., SMILES strings), mechanisms of action, and biological target data, which are entered directly into the database through specialized interfaces. Primary data may also originate from in-house experimental studies, such as investigations into compound–target interactions, which undergo rigorous validation before integration. Additionally, external sources like PubChem or DrugBank [10] can be treated as primary data when manually curated and standardized to align with the database’s criteria. Importantly, primary data has no external references—it is created or adapted exclusively for ChronobioticsDB.

Secondary data, on the other hand, consists of pre-existing information sourced from external publications, databases, or regulatory reports that were originally compiled for purposes unrelated to ChronobioticsDB. Examples include peer-reviewed articles describing compound properties, FDA approval statuses, or datasets from repositories like PubChem and UniProt or RCSB PDB [14]. Unlike primary data, secondary data is linked to external references. There is also strict scrutiny and adaptation to ensure alignment with the database’s standards, involving cross-referencing multiple sources and linking entries to their original references for transparency. Secondary data fulfill the database by providing a broader context.

The synergy between these two types of data is critical: primary data offers specificity and originality tailored to ChronobioticsDB’s objectives, while secondary data expands its scope by integrating the established knowledge. For instance, a compound’s mechanism of action (primary data) might be contextualized using pharmacological profiles from DrugBank [10] (secondary data), creating a multidimensional view of its therapeutic potential. While challenges persist in balancing the dynamic nature of scientific knowledge with the need for stability in the database, this combination ensures that ChronobioticsDB remains a reliable repository and a dynamic tool for researchers exploring circadian rhythm modulation.

### 4.3. Database Architecture

The database architecture of our project is constructed on a relational model and employs PostgreSQL as the database management system (DBMS). The database is organized around the central table, *Chronobiotic*, which stores information pertaining to chemical compounds. The remaining tables are linked to the central table through foreign keys and many-to-many relationships, ensuring both flexibility and data integrity. The database architecture has been designed in accordance with the project’s requirements, including support for both primary and secondary data.

### 4.4. Core Components of the Architecture

#### 4.4.1. Central Table: Chronobiotic

This table serves as the core component and contains primary information about chemical compounds. It includes fields for storing the compound’s name, structure, description, FDA status, references to external resources, and other relevant data (Figure 5).

*Key Features are Unique Fields:* The fields *gname*, *smiles*, *molecula*, and *iupacname* ensure the uniqueness of each compound.

*Relationships with Other Tables:* The *Chronobiotic* table is linked to the *synonyms*, *target*, *mechanism*, and *class* tables through one-to-many and many-to-many relationships.

#### 4.4.2. Auxiliary Tables

The auxiliary tables (*synonyms*, *target*, *mechanism*, and *class*) are utilized to store supplementary information that extends and enriches the data on chemical compounds:
One-to-Many Relationships: The *synonyms* table is connected to *Chronobiotic* via the foreign key *originalbiotic*.Many-to-Many Relationships: The *target*, *mechanism*, *effects*, *article*, and *class* tables are linked to *Chronobiotic* through intermediary tables, which are automatically generated by Django.


#### 4.4.3. Schema of Table Relationships


Chronobiotic → synonyms: A single compound may have multiple synonyms.Chronobiotic → target: A single compound may interact with multiple targets, and a single target may be associated with multiple compounds.Chronobiotic → mechanism: A single compound may exhibit multiple mechanisms of action, and a single mechanism may be associated with multiple compounds.Chronobiotic → class: A single compound may belong to multiple classes, and a single class may encompass multiple compounds.Chronobiotic → effects. A single compound may exhibit multiple effects on circadian rhythms.Chronobiotic → article. A single compound may exhibit multiple literature sources where it is described, and one source also may contain many compounds.


#### 4.4.4. Technologies and Tools


**DBMS:** PostgreSQL, a robust and reliable relational database, provides high performance and supports complex queries [15].**ORM**: Django ORM (Django 5.1.2) is employed for database interactions at the Python (v3.11) code level. This eliminates the need for manual SQL query writing and facilitates efficient data management [16].**Indexes:** Indexes have been created on frequently queried fields, such as *gname*, *smiles*, and *targetsname*, to optimize search performance.**Migrations:** Django’s built-in migration system allows for seamless modifications to the database structure without data loss.


#### 4.4.5. Ensuring Data Integrity and Security


**Foreign Keys:** All inter-table relationships are implemented through foreign keys, ensuring data integrity.**Unique Constraints:** Unique fields (*gname*, *smiles*, *molecula*, and *iupacname*) prevent record duplication.**Role-Based Access Control:** Database access is restricted at the user and role levels, ensuring data security.**Encryption:** Confidential data is stored in an encrypted format.


### 4.5. Use of Artificial Intelligence

Thus far, the assembly and annotation of ChronobioticsDB v1.0 have been conducted manually by the research team to ensure high-quality curation. Moving forward, we recognize the potential of artificial intelligence tools to support this effort. For instance, text-mining algorithms could assist in scanning new literature for possible chronobiotic compounds, and machine learning models might help identify patterns or predict novel chronotherapeutic agents from the existing data. We plan to explore these AI-driven approaches in future updates of ChronobioticsDB while maintaining expert oversight to validate any AI-generated inputs.

## 5. Conclusions and Future Perspectives

ChronobioticsDB represents a pioneering effort to consolidate and organize knowledge on compounds that modulate circadian rhythms. By addressing the fragmentation in chronobiotic research, bridging the gap between chronobiology and bioinformatics, and providing a platform for translational research, this database has the potential to significantly advance our understanding of circadian rhythms and their therapeutic applications. As the field continues to evolve, ChronobioticsDB will serve as a valuable resource for researchers, clinicians, and drug developers, ultimately contributing to the development of novel therapies for circadian-related disorders. Future efforts to expand and enhance the database will further solidify its role as a cornerstone of chronobiological research.

### 5.1. Content Expansion and Data Curation

Our research strategy involves augmenting the existing dataset through the utilization of the Decimer AI platform [17], which facilitates the ML/AI-powered extraction of chemical formulas from primary literature sources, subsequently converting them into standardized existing SMILES notations and IUPAC nomenclature. This process will generate a foundational dataset that is currently unavailable in a machine-readable format across publicly accessible repositories dedicated to chronobiology. By systematically compiling this information, we aim to establish a comprehensive and structured database that enhances the accessibility and utility of chemical data for computational analyses in circadian pharmacology.

### 5.2. Integration of AI-Driven Search and Assisting Tools

A significant advancement under development is the incorporation of a large language model (LLM)-based AI agent (chatGPT, qwen, or deepseek) into the database infrastructure. This integration will be implemented either as an embedded search enhancement or as a standalone AI-assisted directory hosted on an independent server with a strong GPU. The primary objective of this initiative is to optimize query processing and facilitate the automated generation of analytical reports for both developers and end-users. Furthermore, this system will incorporate predictive functionalities to propose potential therapeutic strategies for sleep disorders and desynchronoses of different etiologies, leveraging AI-driven pattern chronobiotic describing text recognition and pharmacological profiling in addition to clinical and translational research.

### 5.3. Computational Chronobiotics Discovery

Expanding upon the existing ChronobioticsDB framework, our team is actively engaged in enriching the dataset to enable advanced computational screening for novel multitarget chronobiotics. Utilizing neural network-based predictive modeling, we intend to establish a new classification of putative chronobiotic compounds. Molecular docking simulations will be employed to prioritize circadian-relevant targets across the entire drug candidate pool, supplemented by inverse docking methodologies to evaluate interactions with key regulatory proteins governing circadian rhythms and potential aging-related pathways. This approach aims to accelerate the identification of compounds with dual chronobiotic and geroprotective properties.

### 5.4. Gerontological Applications and Predictive Modeling

As the dysregulation of circadian rhythms in the aging population is one of the key issues, the final research direction focuses on the geroprotective potential of the database elements. Given the availability of multiple algorithmic frameworks for geroprotector identification within drug datasets, we will maintain synchronization with the latest geroprotector databases or utilize our expanded dataset to test specialized neural networks for geroprotector discovery (the present day dataset includes at least 10 geroprotectors). This will ensure that the database remains a dynamic resource for aging research, capable of identifying compounds with potential lifespan-extending and healthspan-promoting effects through systematic computational screening.

By implementing these initiatives, our project seeks to bridge existing gaps in machine-readable chemical and pharmacological data while establishing a robust, AI-enhanced platform for drug discovery and therapeutic development in chronomedicine and chronogerontology.

## Figures and Tables

**Figure 1 clockssleep-07-00030-f001:**
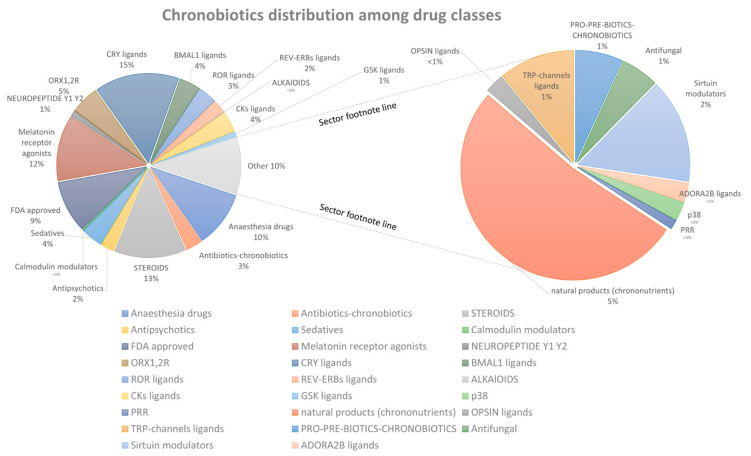
The analysis of ChronobioticsDB content according to pharmacological functional groups and targets.

**Figure 2 clockssleep-07-00030-f002:**
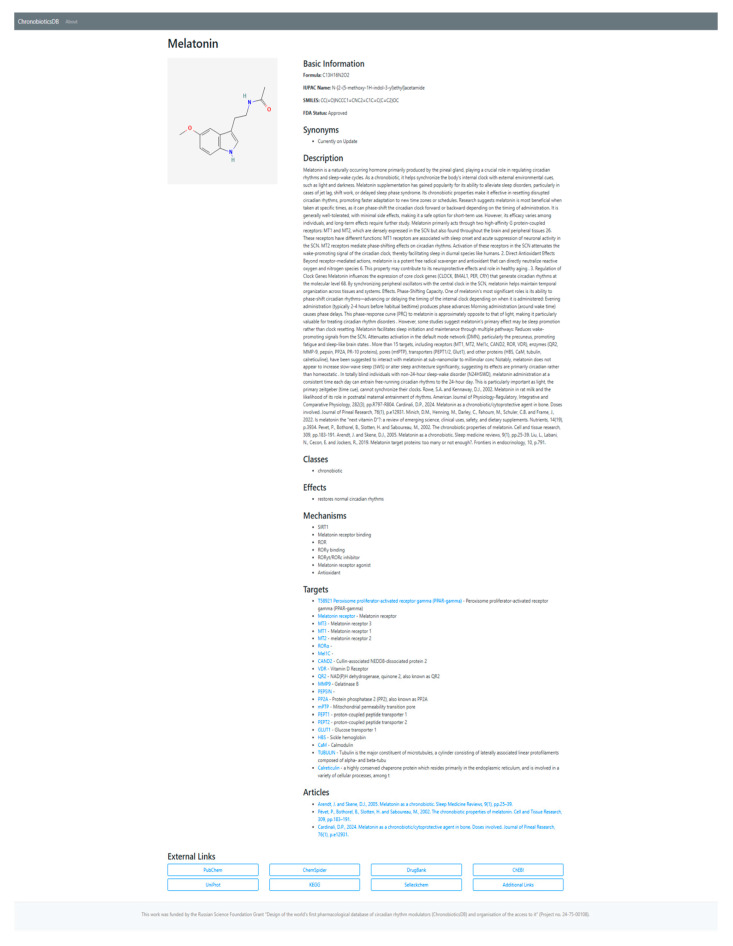
The ChronobioticDB user front-end illustrating the Django-powered interface for a chronobiotic card. By segregating the public and admin interfaces, ChronobioticDB streamlines the contribution process for experts while maintaining an up-to-date, reliable repository that is readily accessible to the broader community for research and reference; nevertheless, the access windows have almost the same information with different rights.

**Figure 3 clockssleep-07-00030-f003:**
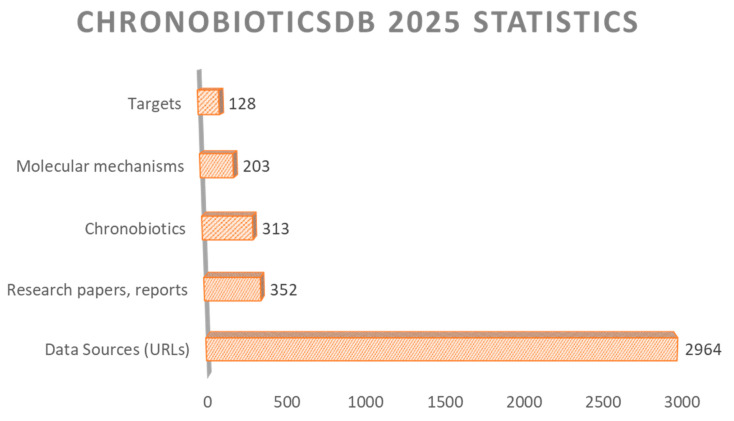
Basic statistics of ChronobioticsDB content.

**Figure 4 clockssleep-07-00030-f004:**
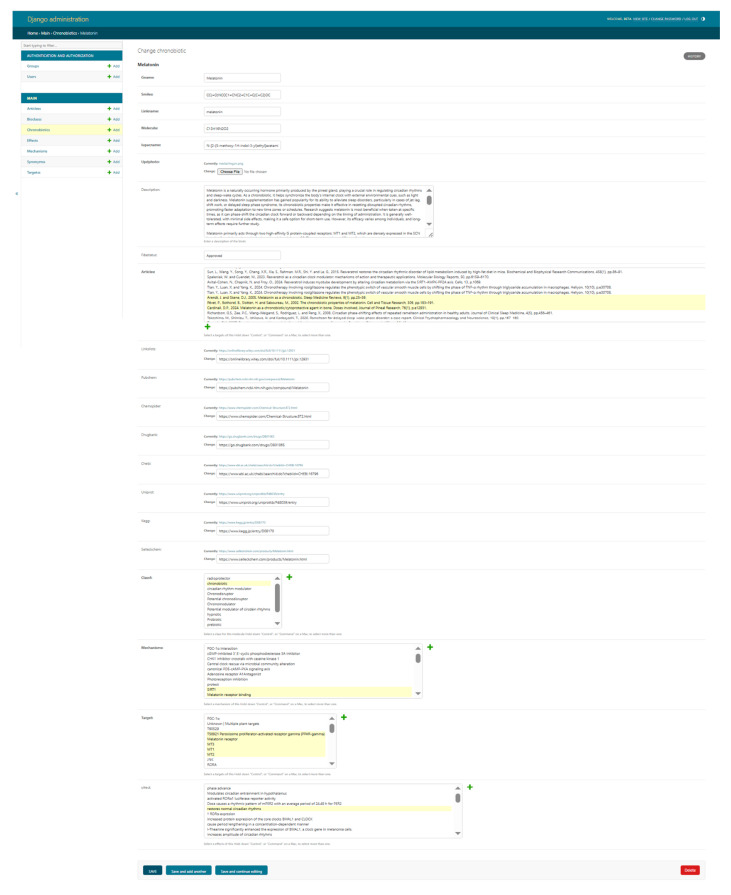
Detailed ChronobioticsDB record for melatonin. The figure displays the administrator-only “Change chronobiotic” form that underpins every compound entry. The header shows breadcrumbs, the object name, and a History link for full audit tracing. The first block contains core chemical identifiers: **Gname** (canonical name and primary key), **SMILES**, URL-friendly **Linkname**, and the molecular formula and full **IUPAC** name, plus an **Updphoto** field for uploading or replacing a structure image. A rich-text **Description** box follows, summarizing the pharmacology, chronobiotic context, and principal findings, while the **Fdastatus** field records regulatory classification (e.g., Approved). It also displays a suite of hyperlink fields and then connects the entry to external resources—**PubMed**, **DrugBank**, **PubChem**, **ChEBI**, **UniProt**, **KEGG**, **ChemSpider,** and any additional literature—ensuring transparent provenance. Functional annotation is captured in three multi-select widgets: **Classf** for broad activity labels (e.g., chronobiotic and hypnotic), **Mechanisms** for curated mechanistic phrases (such as “MT1 receptor binding” or “CK1δ inhibition”), and **Target** for specific molecular entities influenced by the compound (e.g., MT1, MT2, and CRY1). It displays standard Django-admin buttons at the bottom allowing saving, continuing edits, adding another record, or deleting the entry. Together these elements illustrate the depth and editability of metadata stored for each chronobiotic compound in ChronobioticsDB.

**Figure 5 clockssleep-07-00030-f005:**
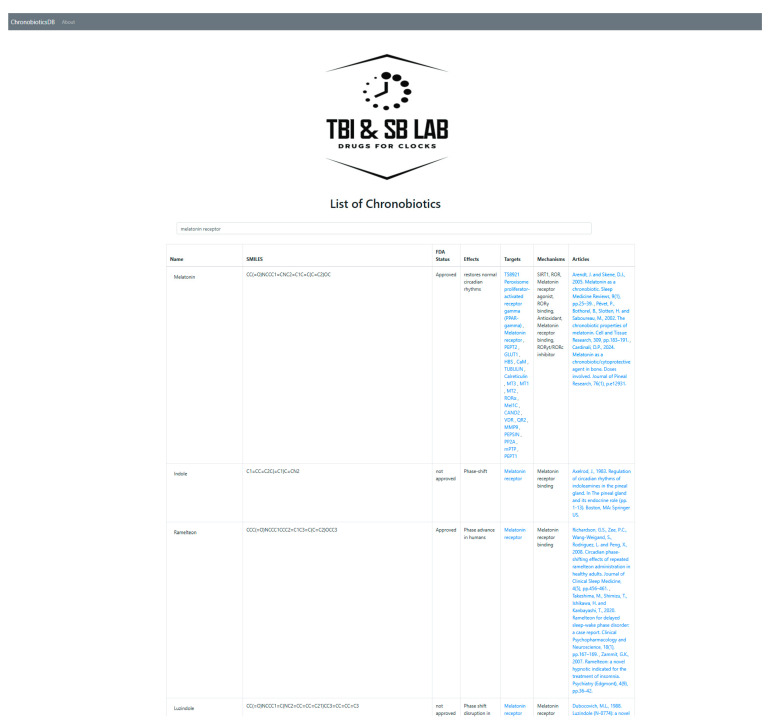
The example of the Search function in the main table of the database using the term of target MT1—melatonin receptor. This image represents both the target-oriented search and the main table of ChronobioticsDB. The search query is applied simultaneously to all columns of the main table.

**Table 1 clockssleep-07-00030-t001:** Criteria of inclusion and exclusion of articles/sources in ChronobioticsDB.

Trait of Research	CRITERIA	
	Research Inclusion	Research Exclusion
Date of Publication	Present	Absent
Age of an article, years	>0	>50
The source/subject is relevant to the research question	Yes	No
Appropriate academic and technical level	Yes	No
Source authority and authorship	Peer-reviewed journal article, scientific report, governmental or academic website, description of authors, presence of affiliation, publisher information	Non-peer-reviewed sources, incomplete information about author, affiliations, publisher, journal or book, or online resource
Accuracy of information presentation in the source	There are no mistakes or unclear statements. Statements are supported by evidence. Information is reliable and is presented in reliable form.	Numerous mistakes, statements unsupported by evidence, unreliable information, and unclear presentation of it.
Purpose of the source publication	Academic or technical use	Entertainment, opinion, propaganda
Cited literature in the source	Bibliography, link in the text with a description of the source	Absence of any links and bibliographic records or inappropriate non-scholarly sources cited
Effect on circadian rhythm described in the article	Present	Absent
Sample size, objects, or patients treated	>30	<30
Reproducibility	Methods are reproducible (Clearly described source of compound or way of extraction/synthesis, doses, regimen, model organism strain or patients cohort described, the method of circadian rhythm measurements and statistics are represented properly)	Not reproducible, speculation (Not clearly described source of compound or way of extraction/synthesis, nonclear doses, regimen. Model organism strain or patient cohort is not appropriate for academic study, the methods of circadian rhythm measurements and statistics are not described or mentioned in general aspect without citation)
Interaction with target	Described	Not described
Model object	Having a circadian molecular clock and circadian rhythms of physiological and molecular processes	The circadian patterns in the object are not described and there is no molecular machinery of the oscillator
Ethical aspect	Ethically appropriate protocol of study, verified with an ethical committee if needed	Illegal or unethical protocol described
Presence of chemical compound or living organism (if probiotic)	Yes	No
Presence of the mechanism of activity	Yes	No
Presence of a chemical graphic formula	Yes	No

## Data Availability

The raw data supporting the conclusions of this article will be made available by the authors on request in the form of an Excel table. Data and code are available at GitHub—YagovkinaA/chronobioticdb (https://github.com/YagovkinaA/chronobioticdb, access date 16 June 2025). The database is hosted at https://chronobiotic.ru/ (access date 16 June 2025).

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
