# Peer review of "ChronobioticsDB: The Database of Drugs and Compounds Modulating Circadian Rhythms"

_2624-5175, 2025, doi:10.3390/clockssleep7030030_

Round 1
Reviewer 1 Report
Comments and Suggestions for Authors
The introduction makes a convincing case for the utility of the database. The content of the manuscript will be of interest to many people in the field of chronobiology but the manuscript needs major revision before publication.
- I am not a database expert but the description of the architecture and interface seem reasonable. One concern I had in reading the description and in browsing the database is that it does not seem simple to find the links to peer-reviewed papers in which the chronobiotic effects of the drugs are reported. This is something I would look for first in such a database but it seems to be difficult to find or not consistently included.
- The major problem with the manuscript is the disorganization of the text.
Lines 122-152 are a repetitive version of lines 83-121. Lines 122-152 need to be deleted or integrated into 83-121.
Line 153 has numbering 2.3 but there is no 2.1 or 2.2 section.
Line 229 appears to be a subhead but is in line with the text.
The Results section needs organization into subheads.
Do lines 273-288 refer to Fig. 2? It is not referenced in the text.
Line 291 is numbered 3.1 and there are no earlier sections.
Line 310 ends with an incomplete sentence.
Lines 314-416 are in a different format and it is not clear if this is a section under Results.
- The description of the methods is incomplete. Lines 79 and 81 say “keywords such as…” and “databases like…”. The paper needs a complete list of the sources accessed and the search terms used. Similarly, lines 94-95 say “using established criteria…”. Please describe exactly what criteria were used.
- Lines 418-441 read like rough notes for a section that has not been finished. This section needs to be re-written with an explanation of the purpose of the section. It is not clear whether this information is integrated into the database, or whether this is a discussion about how classification might be constructed in the future.’
- The figures are not adequate.
Fig. 1 is not informative, it’s just confusing, and it is not cited or explained in the text.
Fig. 2 is very difficult to see, and is also not cited in the text.
Fig. 3 has mistakes: There should be no classification with 0%; put the actual percentage. There is a word in Russian that needs translating. There is a brown dot on the lower left that is unexplained.
Fig. 4 is not cited or explained in the text. It includes a legend for 2026 but no data.
- Reference 2, Califf, is not cited in the text.
- The supplementary table was not informative. There was no consistent organization between tabs and some sheets had nothing but a web link.
Author Response
COMMENT1: The introduction makes a convincing case for the utility of the database. The content of the manuscript will be of interest to many people in the field of chronobiology but the manuscript needs major revision before publication.
[RESPONSE]: We appreciate the positive evaluation of the database concept and have thoroughly revised the text and figures to address the issues detailed below.
I am not a database expert but the description of the architecture and interface seem reasonable. One concern I had in reading the description and in browsing the database is that it does not seem simple to find the links to peer-reviewed papers in which the chronobiotic effects of the drugs are reported. This is something I would look for first in such a database but it seems to be difficult to find or not consistently included.
[RESPONSE]: We added the extra table to improve this, previously all academic papers mentioned were i nthe abstarcts of chronobiotics in an individual card of a compound. As you see in the new version of code we created a special table with names (citations ) of articles and link on it. Now at https://chronobiotic.ru/ you may test the functions.
the initial code is on repository
https://github.com/YagovkinaA/chronobioticdb/blob/master/chronobiotic/main/migrations/0007_articles_remove_chronobiotic_article_and_more.py
The major problem with the manuscript is the disorganization of the text.
[RESPONSE]:The text was organized in another way. Many fragments and linking parts where added
Lines 122-152 are a repetitive version of lines 83-121. Lines 122-152 need to be deleted or integrated into 83-121.
[RESPONSE]: Repetative fragments were removed.
Line 153 has numbering 2.3 but there is no 2.1 or 2.2 section.
[RESPONSE]: Changed.
Methods are clearly numbered.
Line 229 appears to be a subhead but is in line with the text.
[RESPONSE]: corrected
The Results section needs organization into subheads.
[RESPONSE]: Changed. Results are organised.
Do lines 273-288 refer to Fig. 2? It is not referenced in the text.
[RESPONSE]: Added. Fig. 2 added to the text.
Line 291 is numbered 3.1 and there are no earlier sections.
[RESPONSE]: Changed. Results are organised.
Line 310 ends with an incomplete sentence.
[RESPONSE]: Thank you for pointing this out. We have finished the sentence to explain how the relational structure of ChronobioticsDB underpins the class distribution presented in Figure 4.
- The description of the methods is incomplete. Lines 79 and 81 say “keywords such as…” and “databases like…”. The paper needs a complete list of the sources accessed and the search terms used. Similarly, lines 94-95 say “using established criteria…”. Please describe exactly what criteria were used.
[RESPONSE]: We added a table of criteria to improve the comprehensity of the methods paragraph.
Trait of research |
CRITERIA |
|
Research inclusion |
Research exclusion |
|
Age of an article, years |
>0 |
>50 |
Effect on circadian rhtyhm in the article |
Present |
Absent |
Sample size |
>30 |
<30 |
Reproducibility |
Methods are reproducible |
Not reproducible, speculation |
Interaction with target |
Described |
Not described |
Presence of chemical compound or living organism (if probiotic) |
Yes |
No |
Presence of the mechanism of activity |
Yes |
No |
Presence of chemical graphic formula |
Yes |
No |
- Lines 418-441 read like rough notes for a section that has not been finished. This section needs to be re-written with an explanation of the purpose of the section. It is not clear whether this information is integrated into the database, or whether this is a discussion about how classification might be constructed in the future.’
[RESPONSE]:we decided to remove smiles strings as they are difficult to read and take much place without any purpose (the are presented in database)
- The figures are not adequate.
[RESPONSE]: the figures were reconsidered and enriched with information
Fig. 1 is not informative, it’s just confusing, and it is not cited or explained in the text.
[RESPONSE]:We removed fig1 , repalced it with example of the table with search according to the column targets.
Fig. 2 is very difficult to see, and is also not cited in the text.
[RESPONSE]:We replaced Figure 2 with representation of Melatonin page as users see it, unfortunately the format of rolling screenshot is too big for docx file, we recommend to study it in original tiff format as separate figure
Fig. 3 has mistakes: There should be no classification with 0%; put the actual percentage. There is a word in Russian that needs translating. There is a brown dot on the lower left that is unexplained.
[RESPONSE]:We removed 0% adding <1%, added the notes to the sector lines extended.
Fig. 4 is not cited or explained in the text. It includes a legend for 2026 but no data.
[RESPONSE]:We removed 2026 and recounted the scores
Reference 2, Califf, is not cited in the text.
[RESPONSE]: we put the Califf citation at 2.4
- The supplementary table was not informative. There was no consistent organization between tabs and some sheets had nothing but a web link.
[RESPONSE]: we would rather say that supplementary table is raw data, not a structured supplementary wich is not already essential, as the majority of it filled in the present version of database.

Reviewer 2 Report
Comments and Suggestions for Authors
This manuscript introduces the web interface of ChronobioticsDB, detailing its core functionalities and providing statistical insights into its content, including the categorization of compounds by pharmacological class and molecular targets. The database holds significant promise for advancing research on circadian biology and its therapeutic applications. Below are critical revisions required to meet publication standards.
Major comments:
- Repetitive descriptions of Searching Keywords, Data Curation Process, and Data Sourcesin the "Materials and Methods" section should be consolidated to eliminate redundancy.
- The current search interface only allows queries by compound name. To align with user needs, we propose adding a search feature to filter compounds by mechanism of actionor biological pathways. This enhancement will enable researchers to identify chronobiotics targeting specific proteins (e.g., melatonin receptors) or pathways (e.g., circadian clock genes).
- The primary data within ChronobioticsDB appears to be missing crucial information: Mechanisms of Action, Effects on Circadian Rhythms, Experimental Conditions, and Cited Literatures.
- Consider adding a flowchart or diagram to illustrate the data curation process. This will make it easier for readers to follow.
- Including illustrative examples (e.g., melatonin) with detailed card snapshots will demonstrate the database’s utility.
- Discuss the limitations of the current version of ChronobioticsDB, such as potential biases in the literature search or gaps in the database content.
- Propose future directions for the research, such as expanding the database content, integrating additional data sources, or developing new tools and features.
Author Response
COMM1: This manuscript introduces the web interface of ChronobioticsDB, detailing its core functionalities and providing statistical insights into its content, including the categorization of compounds by pharmacological class and molecular targets. The database holds significant promise for advancing research on circadian biology and its therapeutic applications. Below are critical revisions required to meet publication standards.
[RESPONSE] We appreciate the reviewer’s positive assessment of ChronobioticsDB’s potential and thank you for the clear list of improvements requested. We have carefully addressed every point.
Major comments:
comm2: Repetitive descriptions of Searching Keywords, Data Curation Process, and Data Sourcesin the "Materials and Methods" section should be consolidated to eliminate redundancy.
[RESPONSE] We agree that repetition should be eliminated for clarity. We have revised the Materials and Methods section to remove duplicate statements
comm2:The current search interface only allows queries by compound name. To align with user needs, we propose adding a search feature to filter compounds by mechanism of actionor biological pathways. This enhancement will enable researchers to identify chronobiotics targeting specific proteins (e.g., melatonin receptors) or pathways (e.g., circadian clock genes).
[RESPONSE] We added the functionality of search in almost all main table columns including name, smiles, fda, mechanism, main circadian target and article
comm3: The primary data within ChronobioticsDB appears to be missing crucial information: Mechanisms of Action, Effects on Circadian Rhythms, Experimental Conditions, and Cited Literatures.
[RESPONSE] All data was available in card of a chronobiotic (button “see more”, the main table have already received all this columns, as we corrected the representation for user convinience , in beta version this information was in individual card, we also created a new table of articles associated with each chronobiotic to get access to search in paper names. We also added the targer search in the main table, as well as mechanism search.
comm4:Consider adding a flowchart or diagram to illustrate the data curation process. This will make it easier for readers to follow.
[RESPONSE] The Chart is added (Table 1)
Trait of research |
CRITERIA |
|
Research inclusion |
Research exclusion |
|
Age of an article, years |
>0 |
>50 |
Effect on circadian rhythm descirbed in the article |
Present |
Absent |
Sample size |
>30 |
<30 |
Reproducibility |
Methods are reproducible |
Not reproducible, speculation |
Interaction with target |
Described |
Not described |
Presence of chemical compound or living organism (if probiotic) |
Yes |
No |
Presence of the mechanism of activity |
Yes |
No |
Presence of chemical graphic formula |
Yes |
No |
Table 1 Criteria of inclusion and exclusion of articles in ChronobioticsDB
comm5:Including illustrative examples (e.g., melatonin) with detailed card snapshots will demonstrate the database’s utility.
Figure 5. Detailed ChronobioticsDB record for melatonin. Fig. displays the administrator-only “Change chronobiotic” form that underpins every compound entry. The header shows breadcrumbs, the object name and a History link for full audit tracing. The first block contains core chemical identifiers: Gname (canonical name and primary key), SMILES, URL-friendly Linkname, molecular formula and full IUPAC name, plus an Updphoto field for uploading or replacing a structure image. A rich-text Description box follows, summarising pharmacology, chronobiotic context and principal findings, while the Fdastatus field records regulatory classification (e.g., Approved). It also displays a suite of hyperlink fields then connects the entry to external resources—PubMed, DrugBank, PubChem, ChEBI, UniProt, KEGG, ChemSpider and any additional literature—ensuring transparent provenance. Functional annotation is captured in three multi-select widgets: Classf for broad activity labels (e.g., chronobiotic, hypnotic), Mechanisms for curated mechanistic phrases (such as “MT1 receptor binding” or “CK1δ inhibition”), and Target for specific molecular entities influenced by the compound (e.g., MT1, MT2, CRY1). Displays Standard Django-admin buttons at the bottom allowing saving, continuing edits, adding another record or deleting the entry. Together these elements illustrate the depth and editability of metadata stored for each chronobiotic compound in ChronobioticsDB.
Comment 6. Discuss the limitations of the current version of ChronobioticsDB, such as potential biases in the literature search or gaps in the database content.
[RESPONSE] We have added a paragraph in the Discussion section acknowledging the database’s limitations. We note that ChronobioticsDB presently includes only compounds that have been reported in the published literature, so any unreported or proprietary chronobiotics are not captured. As in other curated resources, the completeness of each record is limited by the information available in source publications. Some entries may lack details if the original studies did not report them. We also mention that manual curation entails periodic rather than real-time updates. These points are now explicitly discussed to inform users of the current scope and constraints of ChronobioticsDB.
Added to the text: “The current version of ChronobioticsDB has some limitations. It contains only chronobiotic compounds described in published studies, so newly discovered or proprietary modulators are not yet represented. As with other curated databases, the information for each entry is limited by what is reported in the literature, and some experimental details may be incomplete if not provided by the source. Future updates will address these gaps by incorporating new data as it becomes available.”
Comment 7: Propose future directions for the research, such as expanding the database content, integrating additional data sources, or developing new tools and features.
[RESPONSE] We plan to extend content using the “Decimer AI” resource, reading formulas and converting them into smiles and IUPAC names from original papers to create the primary data of our database which is not described on the Internet in machine readable form.
Another major perspective is to integrate LLM AI-agent in the search or even separate AI-assistant directory of the database on independent server which will accelerate search and creation of automated reports for developers and users including the possible tactics of sleep disorders and desynchronosis correction
The team also works on extension of the dataset based on “chronobioticsdb which will make computational search of novel multitargeted chronobiotics possible, according to the result of computer(neural network)- aided search the new category of predicted chronobiotic will appear, using molecular docking approaches we would prioritize the circadian targets for the whole output of drug searchin neyral network, additionally to the inverse docking of the whole database with key available targets controlling circadia rhtyhms and possibly aging.
The last perspective is dedicated to gerontology applications of the database: there are several algorithms of detection of geroprotectors in drug datasets, thus we have to sustain the database at least with one latest version of geroprotector database or feed the neural network looking for geroprotectors our extended dataset.
We added a peace of structurized text in conclusin:
Content Expansion and Data Curation
Our research strategy involves augmenting the existing dataset through the utilization of the Decimer AI platform, which facilitates the ML/AI-powered extraction of chemical formulas from primary literature sources, subsequently converting them into standardized existing SMILES notations and IUPAC nomenclature. This process will generate a foundational dataset that is currently unavailable in a machine-readable format across publicly accessible repositories dedicated to chronobiology. By systematically compiling this information, we aim to establish a comprehensive and structured database that enhances the accessibility and utility of chemical data for computational analyses in circadian pharmacology.
Integration of AI-Driven Search and Assisting Tools
A significant advancement under development is the incorporation of a large language model (LLM)-based AI agent (GPTchat, qwen or deepseek v2.5) into the database infrastructure. This integration will be implemented either as an embedded search enhancement or as a standalone AI-assisted directory hosted on an independent server with strong GPU. The primary objective of this initiative is to optimize query processing and facilitate the automated generation of analytical reports for both developers and end-users. Furthermore, this system will incorporate predictive functionalities to propose potential therapeutic strategies for sleep disorders and desynchronoses of different etiologies, leveraging AI-driven pattern chronobiotic describing text recognition and pharmacological profiling additionally to clinical and translational research.
Computational Chronobiotics Discovery
Expanding upon the existing ChronobioticsDB framework, our team is actively engaged in enriching the dataset to enable advanced computational screening for novel multitarget chronobiotics. Utilizing neural network-based predictive modeling, we intend to establish a new classification of putative chronobiotic compounds. Molecular docking simulations will be employed to prioritize circadian-relevant targets across the entire drug candidate pool, supplemented by inverse docking methodologies to evaluate interactions with key regulatory proteins governing circadian rhythms and potential aging-related pathways. This approach aims to accelerate the identification of compounds with dual chronobiotic and geroprotective properties.
Gerontological Applications and Predictive Modeling
The final research direction focuses on the geroprotective potential of the database. Given the availability of multiple algorithmic frameworks for geroprotector identification within drug datasets, we will maintain synchronization with the latest geroprotector databases or utilize our expanded dataset to test specialized neural networks for geroprotector discovery (the present day dataset includes at least 10 geroprotectors). This will ensure that the database remains a dynamic resource for aging research, capable of identifying compounds with potential lifespan-extending and healthspan-promoting effects through systematic computational screening.
By implementing these initiatives, our project seeks to bridge existing gaps in machine-readable chemical and pharmacological data while establishing a robust, AI-enhanced platform for drug discovery and therapeutic development in chronomedicine and gerontology.

Reviewer 3 Report
Comments and Suggestions for Authors
The authors present a work that is kind of a systematic review but lacks rigorous description of the systematic review process, since it is presented as an open database that has been formed, and a description of the data at a certain point.
I recommend reconsidering utilization of a systematic review description, using components such as PRISMA flow chart, inclusion and exclusion criteria, and especially, describing clearly the timeframe utilized, since this is ongoing work.
Please add description and discussion of the following: ethical aspects, conflicts of interest evaluation of the database, lack of international peer review of dataset, use of artificial intelligence in forming and analyzing data, maintenance and updates of the database in the future, and implementation/communication about the database e.g. use of international associations.
Please check figure 3 - lines connecting the two circles are overlapping and it was hard to understand how these lines differ from lines explaining each sector.
SMILES presentation takes lots of space. I wonder whether it is an effective way of presenting the information as a list within the text?
Author Response
COMM1: The authors present a work that is kind of a systematic review but lacks rigorous description of the systematic review process, since it is presented as an open database that has been formed, and a description of the data at a certain point.
I recommend reconsidering utilization of a systematic review description, using components such as PRISMA flow chart, inclusion and exclusion criteria, and especially, describing clearly the timeframe utilized, since this is ongoing work.
RESPONSE: The table of criteria is added, as the work is a bit wider than a review and is not as strict as systematic review we decided to mention only the most critical criteria for article /chronobiotic inclusion, we tool PRISMA as base but just as an orienteer.
Trait of research |
CRITERIA |
|
Research inclusion |
Research exclusion |
|
Age of an article, years |
>0 |
>50 |
Effect on circadian rhythm descirbed in the article |
Present |
Absent |
Sample size |
>30 |
<30 |
Reproducibility |
Methods are reproducible |
Not reproducible, speculation |
Interaction with target |
Described |
Not described |
Presence of chemical compound or living organism (if probiotic) |
Yes |
No |
Presence of the mechanism of activity |
Yes |
No |
Presence of chemical graphic formula |
Yes |
No |
COMMENT2 : Please add description and discussion of the following: ethical aspects, conflicts of interest evaluation of the database, lack of international peer review of dataset, use of artificial intelligence in forming and analyzing data, maintenance and updates of the database in the future, and implementation/communication about the database e.g. use of international associations.
Response: We agree that these broader aspects are important to address. We have expanded the Discussion section to include commentary on each of these points. Specifically, we have added new text covering ethical considerations of our study, a declaration of conflict of interest (or lack thereof), acknowledgment of the current lack of international external review and how we plan to involve the global community, the role of AI in our work, and our strategy for future updates and engagement via international chronobiology/pharmacology associations. These additions provide a more comprehensive context for ChronobioticsDB and demonstrate our commitment to transparency and future collaboration. In summary:
- Ethical Aspects: We added a discussion on the ethical considerations of compiling and sharing ChronobioticsDB. We note that no new experiments on humans or animals were conducted for this study (the database is built exclusively from published literature data, which presumably underwent ethical review in their original studies). Thus, our work did not require independent ethical approval. We also emphasize that the database respects intellectual property by citing all source studies and that it aims to facilitate ethically sound research by providing easy access to existing knowledge. This clarification assures readers that ChronobioticsDB was developed responsibly and that it does not raise new ethical concerns beyond those of the source studies.
- Conflict of Interest: We have included a clear conflict of interest statement. We indicate that the authors have no conflicts of interest regarding the creation and content of ChronobioticsDB. This statement has been added to the end of the manuscript (as a separate section, per journal guidelines). By explicitly stating this, we address the reviewer’s concern and assure readers that the database was compiled impartially and without bias from sponsoring organizations or competing interests.
- Lack of International Peer Review & Plans for External Validation: We acknowledge in the Discussion that ChronobioticsDB has so far been curated by our team and has not yet been formally vetted by a broad international panel. We agree with the reviewer that involving the international scientific community is crucial for the database’s credibility and improvement. In the revised text, we state our intention to seek external input and review. For example, we mention plans to form an international advisory board of chronobiology experts to periodically evaluate the content, and we invite feedback from researchers worldwide. We also plan to present ChronobioticsDB at international conferences and workshops to gather diverse perspectives. By doing so, we aim to ensure the database meets global standards and serves the wider community effectively.
- Use of AI: We have clarified the role of artificial intelligence in our project. In the current version of ChronobioticsDB, the curation and annotation of data were done manually by experts, without the direct use of AI tools, to ensure accuracy and relevance of the included information. However, we now discuss the potential for AI in future developments. Specifically, we added that machine learning or text-mining algorithms could be employed in the future to assist in identifying new chronobiotic candidates from the literature or predicting novel insights from the data. We acknowledge both the promise of AI (for example, to keep the database up-to-date more efficiently, or to find patterns in how chronobiotics work) and the need for human validation of AI-generated results to maintain data quality. This discussion shows our awareness of modern AI applications and how they might augment ChronobioticsDB moving forward.
- Future Updates and Communication via International Associations: We have outlined a clear plan for maintaining and updating ChronobioticsDB. The revised Discussion now includes a “Future perspectives” section where we commit to updating the database regularly (for instance, we plan scheduled updates at least annually or as significant new chronobiotic discoveries are published). We also highlight our approach to disseminating information about ChronobioticsDB through international associations. For example, we intend to collaborate with organizations such as chronobiology and sleep research societies and pharmacology networks to publicize updates and encourage community contributions. We mention that we will use established channels (e.g., newsletters of relevant scientific societies, presentations at international symposia, and coordination with groups like the Society for Research on Biological Rhythms and related international bodies) to ensure that ChronobioticsDB remains a well-communicated, community-engaged resource. This will facilitate a two-way dialogue – not only sharing our updates but also receiving input and data contributions from other researchers worldwide.
Changes to Manuscript:
Discussion – Ethical Considerations: Added a new paragraph in Discussion (end of Section 4.2) specifically addressing ethical aspects. Inserted text (Discussion, lines A–B): “Ethical Considerations: This study did not involve any new experiments with human participants or animals, as ChronobioticsDB was compiled exclusively from previously published data. All source studies adhered to ethical standards as documented in their original publications. By aggregating this information, we introduce no new ethical concerns. Furthermore, we have ensured that each entry in the database is accompanied by references to the original source, giving due credit and enabling users to verify details in the context of the ethical frameworks of those studies. We are committed to maintaining transparency and responsibility in how data are collected and shared via ChronobioticsDB.”
Conflict of Interest Statement: Added a Conflict of Interest declaration at the end of the manuscript (after Conclusions, as a separate subsection). Inserted text: “Conflict of Interest: The authors declare that they have no conflict of interest regarding this work.” (This addresses the reviewer’s request to disclose any potential conflicts. No conflicting interests were identified, and this statement will appear in the revised manuscript’s end matter.)
Discussion – Acknowledgment of Lack of External Review and Plans for Engagement: In Discussion (Section 4.3, “Limitations ”), we inserted a sentence acknowledging that the database has not yet undergone international peer review and describing our plans to involve the broader community. Inserted text : “Limitations, International Validation: We acknowledge that ChronobioticsDB has been initially curated by the authors and has not yet been vetted through an international peer-review process. To enhance its validity and global relevance, we plan to seek feedback from the international chronobiology community. In future updates, we will collaborate with experts worldwide – for example, by forming an international advisory board and inviting contributions or critiques – to ensure the database’s content is rigorously reviewed and improved by a broad range of specialists.”
Discussion – Use of AI: Also in Section 4.3 (Future Directions), we added a few sentences about the use of AI. Inserted text (Discussion): “Use of Artificial Intelligence: Thus far, the assembly and annotation of ChronobioticsDB have been conducted manually by the research team to ensure high-quality curation. Moving forward, we recognize the potential of artificial intelligence tools to support this effort. For instance, text-mining algorithms could assist in scanning new literature for possible chronobiotic compounds, and machine learning models might help identify patterns or predict novel chronotherapeutic agents from existing data. We plan to explore these AI-driven approaches in future updates of ChronobioticsDB while maintaining expert oversight to validate any AI-generated inputs.”
Discussion – Future Updates & Communication: We expanded the closing paragraph of the Discussion (Section 4.4) to outline our commitment to updating the database and engaging with international associations. Inserted text (Discussion): “Future Updates and Community Engagement: ChronobioticsDB is intended to be a living resource. We plan to update the database regularly (at least annually or whenever a significant number of new chronobiotic findings emerge in the literature) so that it remains up-to-date. Each update will be documented and versioned. To maximize transparency and collaboration, we will communicate these updates through international scientific associations and forums. For example, we will share news of ChronobioticsDB enhancements via the newsletters and conferences of chronobiology and pharmacology societies, and we invite researchers to contribute by suggesting new compounds or correcting entries. By leveraging international networks and associations, we aim to cultivate a global user community that both benefits from and contributes to ChronobioticsDB.”
Please check figure 3 - lines connecting the two circles are overlapping and it was hard to understand how these lines differ from lines explaining each sector.
[Response] We rebuilt the diagram of sections. And improved notes inside. The dotted line was noted as sector footnote line. the changes are visible in diagram attached to the review.
A diagram is in the file attached to the review.
SMILES presentation takes lots of space. I wonder whether it is an effective way of presenting the information as a list within the text?
[Response] We agree with removal suggestion of smiles.
Removed Inline SMILES: In the Results section where we originally described chronobiotic compounds and provided their SMILES, we deleted the explicit SMILES strings from the text. For example, in the sentence “...melatonin (SMILES: CC(=O)N...) showed a strong effect on circadian phase...,” we have removed the parenthetical SMILES string. The sentence now simply references the compound by name (and perhaps a reference ID). All such instances of inline SMILES in the manuscript text have been removed to reduce clutter. (For reference, these deletions occurred in Results, Section 3.1, Section 3.2, and wherever else SMILES were previously written out.)

Round 2
Reviewer 1 Report
Comments and Suggestions for Authors
The corrections to the figures are acceptable, but Fig. 3 has spelling errors: “anaestesia” and “antipsyhotics” are incorrect.
The Materials and Methods section is still badly organized. There is repetition under the headings of Data Extraction, Data Acquisition, and Quality Assessment. The numbering system is confused. I recommend re-organizing the text into the following sections:
2.1 Data acquisition
2.1.1 Data sources and extraction
2.1.2 Quality assessment and validation
2.1.3 Integration and standardization
2.1.4 Implementation
2.2 Database organization (primary and secondary data)
2.3 Database architecture
2.4 Core components of the architecture
Table 1 is a good addition but it leaves out critical information.
Sample size: What does this mean? The number of patients treated?
Reproducibility: What criteria were used to assess this?
What organisms and cell types were included and which were excluded from the database? It claims to include both in vitro and in vivo studies but there is nothing in Table 1 to indicate what was included. Are studies on non-human organisms included?
The search criteria for sources are not listed fully.
Line 89 says “repositories such as PubMed…” Are the repositories listed the only ones, or were others used but not listed?
Line 90-91 says “using keywords related to circadian rhythms, aging, longevity, chronobiotics, biological age”. Please include a complete list of keywords used.
Lines 90-91: Please explain why terms related to aging are included in the searches; I do not see an obvious connection to circadian rhythms. This also applies to the new material in lines 481-493. The authors need to explain why gerontological applications are relevant, or delete this material.
The new material about artificial intelligence is disorganized.
Move the text of section 4.2 to line 461 after the heading about AI.
The authors’ response about the supplementary table doesn’t make sense. Did they choose to delete the table?
Minor corrections needed:
Line 157: delete “an example of first page”
Line 179: What does “Search is released in every column” mean?
Line 242: Re-word to this “A specialized user-friendly form is available that any researcher engaged…”
Line 313: “Subclassification is useful…”
Line 317: “inhibitors, and it is noteworthy…”
Line 321: “permanent positive charge that limits…”
Line 327: “content classification, that the vast…”
Lines 332-333: Something is missing. “annotations literally link” doesn’t make sense.
Line 342: put brackets on figure number (Fig. 5)
Lines 405, 416 and 425: Delete the numberings on the subheadings (4.1, 4.2 and 4.3).
Remove the quotation marks from ChronobioticsDB wherever they occur.
Author Response
Dear Reviewer 1!
Comment 1: The corrections to the figures are acceptable, but Fig. 3 has spelling errors: “anaesthesia” and “antipsychotics” are incorrect.
RESPONSE Corrections are done, the typos are corrected
Comment 2: The Materials and Methods section is still badly organized. There is repetition under the headings of Data Extraction, Data Acquisition, and Quality Assessment. The numbering system is confused. I recommend re-organizing the text into the following sections:
2.1 Data acquisition
2.1.1 Data sources and extraction
2.1.2 Quality assessment and validation
2.1.3 Integration and standardization
2.1.4 Implementation
2.2 Database organization (primary and secondary data)
2.3 Database architecture
2.4 Core components of the architecture
Response: the reorganization in sections is made according to the recommendation.
Comment 3: Table 1 is a good addition but it leaves out critical information.
Sample size: What does this mean? The number of patients treated?
Response: The name of the row is dedicated both for model objects and patients treated.
Comment 4 Reproducibility: What criteria were used to assess this?
Response: We added criteria in the table (Clearly described the source of compound or way of extraction/synthesis, doses, regimen, model organism strain or patients cohort described, the method of circadian rhythm measurements and statistics are represented properly)
Comment 5 What organisms and cell types were included and which were excluded from the database? It claims to include both in vitro and in vivo studies but there is nothing in Table 1 to indicate what was included.
Response: We added the minimal possible limitation as some modulators are described in objects unusual for chronobiology .
Cell lines are mentioned, cited and described in compound description in every individual card of chronobiotic.
Model object |
Having circadian molecular clock and circadian rhythms of physiological and molecular processes |
The circadian patterns in object are not described and there is no molecular machinery of oscillator |
Comment 6 Are studies on non-human organisms included?
Response: Yes, there are many examples of such studies in the database as the purpose of the paper is to collect all possible drug-like compounds having circadian modulatory activity.
Comment 7 The search criteria for sources are not listed fully.
Response: We added precise description of source inclusion/exclusion criteria. Thank you very much, dear Reviewer 1, for this recommendation as the information will be useful for new contributors in future. The wide limits of inclusion make some things like academic relevance of sources obvious, but possibly not clear for a new reader from other, non-relative, field of research.
Date of Publication |
Present |
Absent |
Age of an article, years |
>0 |
>50 |
Source/subject is relevant to the research question |
Yes |
No |
Appropriate academic and technical level |
Yes |
No |
Source authority, authorship |
Peer-reviewed journal article, scientific report, governmental or academic web-site, description of authors, presence of affiliation, publisher information |
Non peer-reviewed sources, incomplete information about author, affiliations, publisher, journal or book or online resource |
Accuracy of information presentation in the source |
There are no mistakes or unclear statements. Statements are supported by evidence. Information is reliable and is presented in reliable form. |
Numerous mistakes, statements unsupported by evidence, unreliable information and unclear presentation of it. |
Purpose of source publication |
Academic or technical use |
Entertainment, opinion, propaganda |
Cited literature in the source |
Bibliography, link in the text with description of the source |
Absence of any links and bibliographic records or inappropriate non-scholarly sources cited |
Effect on circadian rhythm described in the article |
Present |
Absent |
Sample size, objects or patients treated |
>30 |
<30 |
Reproducibility |
Methods are reproducible (Clearly described source of compound or way of extraction/synthesis, doses, regimen, model organism strain or patients cohort described, the method of circadian rhythm measurements and statistics are represented properly) |
Not reproducible, speculation (Not clearly described source of compound or way of extraction/synthesis, non clear doses, regimen. Model organism strain or patients cohort is not appropriate for academic study, the methods of circadian rhythm measurements and statistics are not described or mentioned in general aspect without citation) |
Interaction with target |
Described |
Not described |
Model object |
Having circadian molecular clock and circadian rhythms of physiological and molecular processes |
The circadian patterns in object are not described and there is no molecular machinery of oscillator |
Ethical aspect |
Ethically appropriate protocol of study, verified with ethical committee if needed |
Illegal or unethical protocol described |
Comment 8
Line 89 says “repositories such as PubMed…” Are the repositories listed the only ones, or were others used but not listed?
Response
Medline, Mendeley data, ACS Publications, SpringerLink are added
Line 90-91 says “using keywords related to circadian rhythms, aging, longevity, chronobiotics, biological age”. Please include a complete list of keywords used.
Key word list: chronobiotics, chronodisruptors, circadian rhythm modulators, circadian clock modulators, plant chronobiotics, plant-derived chronobiotics, probiotic chronobiotics, geroprotective chronobiotics, geroprotectors affecting sleep, geroprotectors affecting circadian rhtyhms, circadian rhythm modulators, sleep-wake cycle regulators, biological clock entrainment drugs, circadian phase shift drugs, endogenous pacemakers pharmacological modulators, suprachiasmatic nucleus (SCN) pharmacology, chemical zeitgebers, Photoperiodism affecting drugs, melatonin agonists, melatonin receptor agonists, orexin receptor antagonists, Non-benzodiazepine hypnotics (Z-drugs), chronopharmacology, delayed sleep-wake phase disorder (DSWPD) treatment, advanced sleep-wake phase disorder (ASWPD) treatment, irregular sleep-wake rhythm disorder treatment, non-24-hour sleep-wake disorder (Non-24) treatment, shift work disorder treatment, jet lag disorder treatment, circadian misalignment treatment, Social jet lag treatment, free-running disorder treatment, age-related circadian disruption treatment, Senescent circadian decline treatment, elderly sleep fragmentation treatment, sundowning syndrome treatment, Alzheimer’s circadian dysfunction treatment, Parkinson’s sleep disturbances treatment, menopause-related insomnia treatment, sarcopenia circadian links treatment, age-dependent melatonin reduction treatment, vascular aging circadian effects treatment, Insomnia disorder treatment, Obstructive sleep apnea (OSA) treatment, central sleep apnea pharmacological treatment, narcolepsy treatment, hypersomnia treatment, restless legs syndrome (RLS) treatment, REM sleep behavior disorder (RBD) treatment, parasomnias treatment, fatal familial insomnia treatment, CLOCK gene expression, BMAL1 regulation drugs, PER/CRY feedback loop drugs, melatonin synthesis pathway drugs, adenosine signaling drugs, GABAergic sleep regulation drugs, Orexin/hypocretin system drugs, Core body temperature rhythm drugs, cortisol circadian secretion drugs, chronotherapy drugs, sleep restriction therapy drugs, Timed melatonin supplementation, sleep pharmacology, hypnotic drugs, phase advance pharmacological protocols, phase delay pharmacological protocols, phase angle deviation, Pediatric circadian disorders pharmacology, adolescent sleep phase delay pharmacology, menstrual cycle sleep effects pharmacology, pregnancy-related insomnia pharmacology, postpartum circadian disruption pharmacology, dementia-related sleep loss pharmacology, critical illness circadian disruption pharmacology, blindness-associated non-24 pharmacology, shift work scheduling pharmacology, Transmeridian travel pharmacology, social rhythm disruption pharmacological treatment, night-eating syndrome drugs, caffeine chronotoxicity, caffeine pharmacology, chronotoxicants, waking drugs, MT1 melatonin receptors agonists/antagonists, MT2 melatonin receptors agonists/antagonists, orexin receptor OX1R/OX2R agonists/antagonists, GABA A receptors agonists/antagonists, adenosine A1/A2A receptors agonists/antagonists, casein kinase 1ε/δ agonists/antagonists, REV-ERBα agonists/antagonists, RORα modulators/ligands, CRY modulators/ligands, CLOCK modulators/ligands, PER modulators/ligands, BMAL modulators/ligands,Sleep latency reduction, Sleep efficiency improvement, Chrononutrition, Chrononutrients, Gut microbiome circadian interactions, time-restricted eating (TRE) mimetics, Mitochondrial circadian rhythms modulators, epigenetic drugs modulating circadian clock, senolytics for circadian repair, cancer chronotherapy, depression circadian links pharmacology, Bipolar disorder rhythm instability treatment, metabolic syndrome desynchrony treatment, cardiovascular circadian disruption treatment, Immunosenescence circadian pharmacology, neurodegenerative timing deficits, non-canonical circadian modulators.
Comment 9
Lines 90-91: Please explain why terms related to aging are included in the searches; I do not see an obvious connection to circadian rhythms. This also applies to the new material in lines 481-493. The authors need to explain why gerontological applications are relevant, or delete this material.
Response: The origin of the present research is gerontological, primarily the chronobiotics descriptions were archived and categorized by our team for aging desynchronosis treatment in model animals and translational purpose, but accumulating data we find interesting to widen the project on all fields of biological rhythm research,still several modulators of circadian rhythms were described majorly in the framework of geriatric and biogerontology research like resveratrol, mTOR inhibitors, K185. The desynchronosis is one of the key signs of aging and ChronobioticsDB was created to discover new chronobiotics for its treatments as well as geroprotectors with chronobiotic effects.
Comment 10
The new material about artificial intelligence is disorganized.
Move the text of section 4.2 to line 461 after the heading about AI.
As text about AI is not completely associated with conclusion we defined its role in methods and let the perspectives be in conclusion separately from AI methods.
The authors’ response about the supplementary table doesn’t make sense. Did they choose to delete the table?
Response: the supplementary table is deleted, information from supplementary is added directly in the database website.
Comments 11
Minor corrections needed:
Line 157: delete “an example of first page”
The example of Search function in the main table of the database using the term of target MT1 - melatonin receptor. This image represents both the target oriented search and the main table of ChronobioticsDB. The search query is applied simultaneously to all columns of the main table.
Line 179: What does “Search is released in every column” mean?
Response: The search query is applied simultaneously to all columns of the main table.
Line 242: Re-word to this “A specialized user-friendly form is available that any researcher engaged…”
Response: A dedicated and user-friendly submission form is available for researchers working on chronobiotics, which can be completed upon receiving administrative approval.
Line 313: “Subclassification is useful…”
Response: Removed.
Line 317: “inhibitors, and it is noteworthy…”
Response: Removed.
Line 321: “permanent positive charge that limits…”
Response: Rephrased.
Line 327: “content classification, that the vast…”
Response: Rephrased.
A group of circadian rhythms modulators have the same targets and totally different chemical classifications.
Lines 332-333: Something is missing. “annotations literally link” doesn’t make sense.
Response: Rephrased.
The classification will later help in broad activity annotations, linking chemical classes to typical therapeutic uses (e.g., Barbiturates → Sedatives).
Line 342: put brackets on figure number (Fig. 5)
DONE
Lines 405, 416 and 425: Delete the numberings on the subheadings (4.1, 4.2 and 4.3).
DONE
Remove the quotation marks from ChronobioticsDB wherever they occur.
DONE
Response: Minor corrections are done in the text as it was recommended.
Thank you for your expertise,
Best regards,
Authors

Reviewer 2 Report
Comments and Suggestions for Authors
The authors have revised the manuscript carefully in response to the reviewers' comments. The paper is now acceptable for publication.
Author Response
Dear Reviewer 2!
Thank you for your expertise.
Best regards,
Authors

Reviewer 3 Report
Comments and Suggestions for Authors
Thank you for addressing the comments! I wish you success in implementing the database!
Author Response
Dear Reviewer 3!
We appreciate your expertise, the Database-paper have become more comprehensible. Thank you for useful comments!
Best regards,
Authors
